# Real-Time Monitoring of the Temperature, Flow, and Pressure Inside High-Temperature Proton Exchange Membrane Fuel Cells

**DOI:** 10.3390/mi13071040

**Published:** 2022-06-30

**Authors:** Chi-Yuan Lee, Fang-Bor Weng, Chun-Wei Chiu, Shubham-Manoj Nawale, Bo-Jui Lai

**Affiliations:** Department of Mechanical Engineering, Yuan Ze Fuel Cell Center, Yuan Ze University, Taoyuan 32003, Taiwan; fangbor@saturn.yzu.edu.tw (F.-B.W.); aa487156aa@gmail.com (C.-W.C.); shubham.nawale86@gmail.com (S.-M.N.); a0928021475@gmail.com (B.-J.L.)

**Keywords:** high-temperature proton exchange membrane fuel cell, three-in-one flexible micro-sensor, real-time micro-monitoring

## Abstract

The proton exchange membrane fuel cell (PEMFC) system is a highly efficient and environmentally friendly energy conversion technology. However, the local temperature, flow, and pressure inhomogeneity within the fuel cell during the electrochemical reaction process may lead to depletion of PEMFC material and uneven fuel distribution, thus affecting the performance and service life of high-temperature PEMFCs. In this study, micromachining technology is used to develop a three-in-one flexible micro-sensor that is resistant to a high-temperature electrochemical environment (120~200 °C). Appropriate materials and process parameters are used to protect the micro-sensor from failure or damage under long-term testing, and to conduct a real-time micro-monitor of the local temperature, flow, and pressure distribution inside high-temperature PEMFCs.

## 1. Introduction

Fuel cells are important electrochemical devices that can convert chemical energy into electrical energy through oxidation–reduction reactions [1,2,3]. They have the advantages of high efficiency and low pollution [4,5]. Especially, the use of polymer electrolytes to separate the anode and cathode proton exchange membrane fuel cells has been applied in many scientific and engineering fields.

The demand for efficient and clean energy is increasing due to the depletion and pollution of fossil fuels. PEMFCs have gradually attracted attention due to their good conversion efficiency (45–50%), environmentally friendly properties, simple structure, and low noise [6].

Fuel cells have been widely used as clean and efficient energy conversion devices, including automobiles, stationary power sources, distributed power generation, and aerospace auxiliary power sources [7]. Although the development potential of fuel cells has received much attention, the problem of the cost must be solved now. At present, fuel cell technology is mainly based on low-temperature PEMFCs. However, the bottlenecks of low-temperature fuel cells include: (1) that anode catalysts have poor resistance to CO poisoning in a low-temperature (<80 °C) environment; (2) the perfluorofluoric acid membrane needs to be in a high-humidity environment for good ionic conductivity; (3) cathodic reduction overpotential is high; and (4) problems in liquid water and heat removal management lead to a slowdown in the time course of mass production. Therefore, international trends are gradually developing towards high-temperature fuel cell technology. The problems faced by high-temperature fuel cells are internal local temperature, flow, and pressure inhomogeneity, which accelerate the aging of the membrane electrode assembly (MEA) membrane materials and lead to the serious degradation of fuel cell performance. Nam et al. [8] successfully developed a patterned mesoporous TiO_2_ microplate (PTMP) (50/500 μm in diameter and ∼5.2 μm in thickness) using a polymer microporous template and a well-controlled auto-spacy technique. Nafion^®^ composite membrane. The MEA under high temperature and low relative humidity conditions (35% RH @ 120 °C) exhibited a maximum power density over 35.2% higher than the reference MEA. Askaripour et al. [9] developed a single PEMFC model capable of predicting the distribution of parameters (temperature, humidity, pressure, cell current density) along the anode and cathode flow channels and analyzing the relationship of parameters to cell performance. Liu [10] et al. studied the effect of flow channel placement direction on the performance of PEMFCs with dead-end anodes (DEAs) under a gravity environment, and found that different flow channel placement directions also had a significant impact on the local current density distribution of PEMFCs and DEAs. Yang et al. [11] used platinum as the thermistor deposited on the membrane surface and calculated the surface temperature with the help of resistance temperature calibration data. Sun et al. [12] used micromachining technology to create a flow sensor and studied the effect of different geometric parameters on the temperature difference. Barnoon et al. [13] examined the mechanical and thermal properties of bipolar plates used in PEMFC under different environmental conditions. Peng et al. [14] developed a 3D transient computational fluid dynamics model for dead-end anode PEMFCs; the effects of various functional parameters in the fuel cell on the cell performance and the resulting current density, the mass fraction of hydrogen and nitrogen, and the time-dependent variation of the volume fraction were investigated in the presence and absence of the purification mechanism. Hosseini et al. [15] used an agglomeration model to numerically analyze 2D single cells and open-anode PEMFCs with water and air inputs; the effects of various parameters such as the stoichiometric coefficient, the saturated water content in the cathode gas diffusion layer, operating temperature and pressure, and relative humidity on fuel cell performance were investigated.

At present, the study on high-temperature PEMFCs is mainly focused on PEMFCs, while the study on flow channel geometry and operating conditions is mainly based on low-temperature PEMFCs. Because the principles of low-temperature PEMFCs and high-temperature PEMFCs are the same, the study and operating conditions of the flow channel can be extended from low-temperature PEMFCs to high-temperature PEMFCs [16,17]. Jo et al. [18] studied the cell performance under different conditions through a numerical model of high-temperature PEMFCs and proved that temperature and air–fuel ratio have important effects on cell performance. Thomas et al. [19] demonstrated the superiority of the newly developed flow field in terms of mass transfer through a numerical model of high-temperature PEMFCs. Wu et al. [20] used a numerical model of a high-temperature PEMFC to study the effect of channel rectangular ribs and obtain the best performance.

The purpose of this study is to use micromachining technology to develop a three-in-one flexible micro-sensor for high-temperature electrochemical environments (120~200 °C) to measure temperature, pressure and flow. The aim is to monitor three important physical quantities in real time and provide actual information about the interior of high-temperature PEMFCs to improve their performance and extend their service life.

## 2. Process of Three-in-One Flexible Micro-Sensors

The flexible substrate used in this study was polyimide (PI) which has advantages such as high temperature resistance (<400 °C), compression resistance, high flexibility, and good durability. The process and optical microscope photos of the three-in-one flexible micro-sensor developed by micromachining technology are shown in Figure 1 and Figure 2. The three-in-one flexible micro-sensor with small size and high sensitivity can be embedded in the interior for real-time micro-diagnosis and analysis without affecting the operation of high-temperature PEMFCs.

## 3. Real-Time Micro-Monitoring of the Three-in-One Flexible Micro-Sensor Embedded in a High-Temperature PEMFC

With high-temperature fuel cell test machines and NI data acquisition devices, the internal information acquisition and micro-diagnosis and analyses of high-temperature fuel cells were carried out. Under the condition of a constant current, the local temperature flow and the pressure changes and distribution in a high-temperature fuel cell were monitored and discussed. The reaction area of the high-temperature fuel cell was 31.4 cm^2^. Under the operating temperature of 160 °C, the different unhumidified air flow rate given to the anode flow rate (H_2_) was 2 lspm and the cathode flow rate (Air) was 4 lspm, and a condition of constant current (0.8 A/cm^2^) was given. By means of the NI PXI 2575 data acquisition device, the local physical quantities in the high-temperature fuel cell were obtained, and then discussed and analyzed. The detailed operating conditions are shown in Table 1. The observed temperature, flow, and pressure changes are shown in Figure 3, Figure 4, Figure 5 and Figure 6.

The temperature change diagram shows that the temperature of the inlet was lower than the outlet, which the gas heated when passing through the flow channel. The heat dissipation at the outlet was poor. Thus, the temperature of outlet was higher than the inlet. In the flow chart, the quality of the flow channel design can be judged according to the difference between the outlet and the inlet. Assuming that the outlet flow rate was smaller than the inlet flow rate, this means that the internal air tightness was not well executed or the flow channel design was poor, resulting in a reduction in the flow rate. These changes will greatly affect the performance of the battery, so the monitoring data of the micro-flow sensor are important indicators for improving the design of the flow channel. Pressure is also an important physical quantity for PEMFC. When the external gas is continuously supplied into the battery, the battery pressure change should be greater than 0. However, if there is air leakage inside the battery, the change in the internal pressure will remain unchanged.

## 4. Conclusions

This study has successfully developed a three-in-one flexible micro-sensor with a small size and high sensitivity. It can be embedded in the interior of high-temperature PEMFCs for real-time micro-diagnoses and analyses of temperature, flow, and pressure distribution without affecting their operation.

## Figures and Tables

**Figure 1 micromachines-13-01040-f001:**
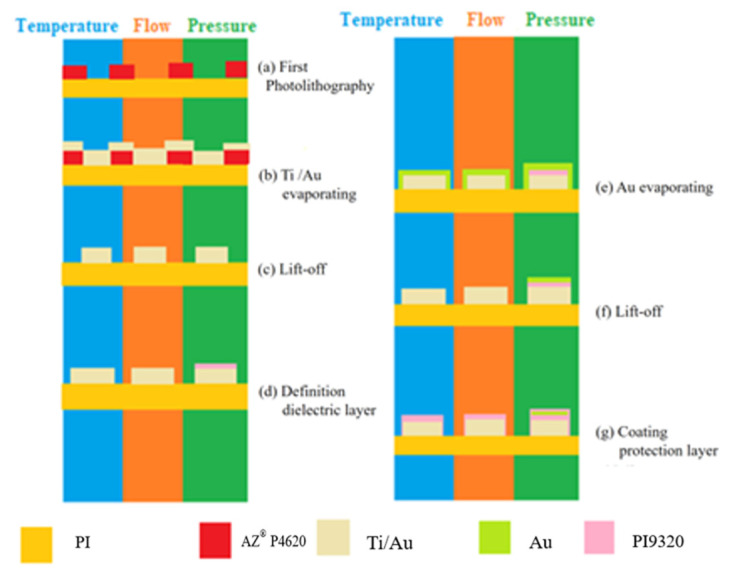
Process diagram of a three-in-one flexible micro-sensor.

**Figure 2 micromachines-13-01040-f002:**
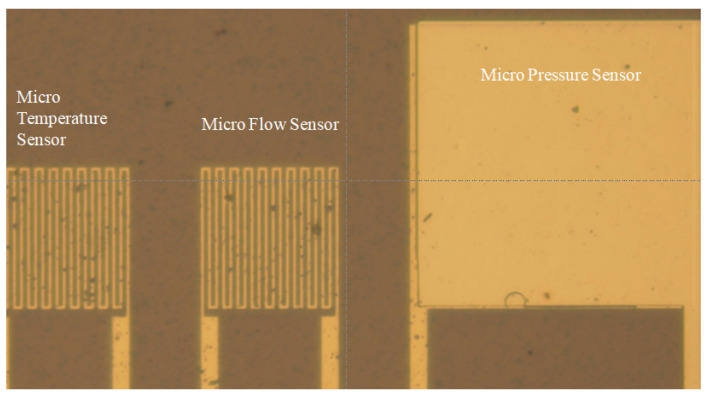
Optical microscope image of a three-in-one flexible micro-sensor.

**Figure 3 micromachines-13-01040-f003:**
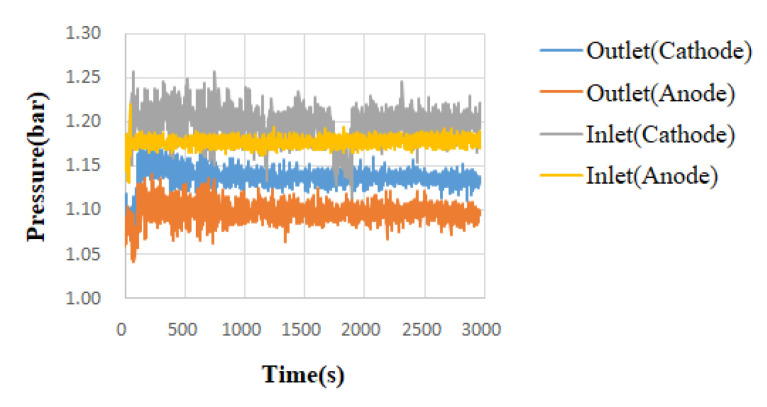
Temperature variation chart of different positions of high-temperature fuel cells.

**Figure 4 micromachines-13-01040-f004:**
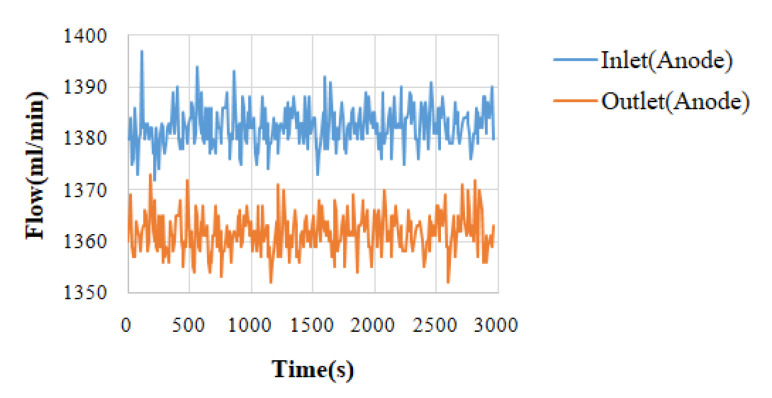
Flow chart of high-temperature fuel cell inlet and outlet (anode side).

**Figure 5 micromachines-13-01040-f005:**
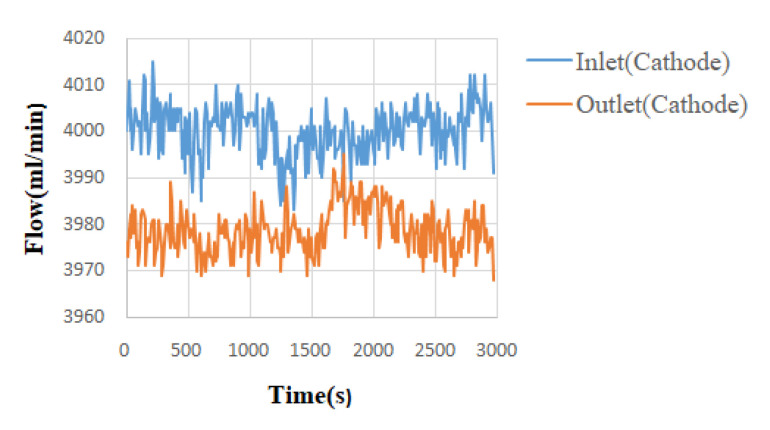
Flow chart of high-temperature fuel cell inlet and outlet (cathode side).

**Figure 6 micromachines-13-01040-f006:**
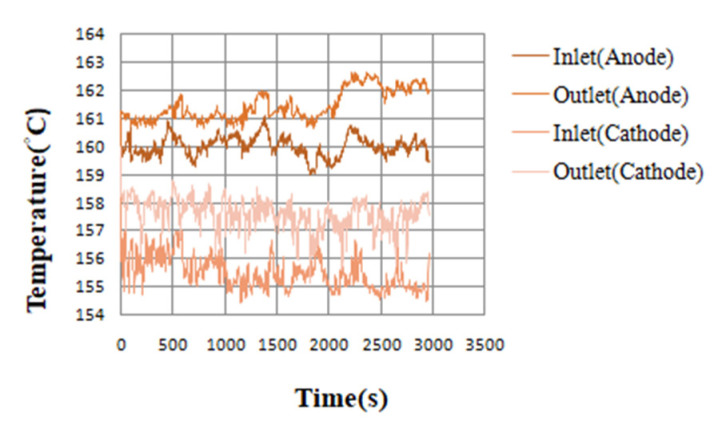
Pressure variation chart of different positions of the high-temperature fuel cell.

**Table 1 micromachines-13-01040-t001:** Test conditions of the high-temperature fuel cell.

Item	Condition
Cell temperature (°C)	160
Anode flow (H_2_) (lspm)	2
Cathode flow (Air) (lspm)	4
Gas temperature	Room temperature
Constant current (A/cm^2^)	0.8
Reaction area (cm^2^)	31.4

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
