# Peer review of "Real-Time Monitoring of the Temperature, Flow, and Pressure Inside High-Temperature Proton Exchange Membrane Fuel Cells"

_micromachines, 2022, doi:10.3390/mi13071040_

Round 1

Reviewer 1 Report

·                  Please clearly put forward the novelty of this paper in the text.

·             A major issue in this paper is that it only outlines data and results, with very little in-depth discussion on the reason behind these data.

·              

·             I think the presentation is not in its best. Authors should re-organize the whole paper and describe what they have done in a clear manner.

·             The following references are related to the paper, and the author is recommended to add them:Energy analysis of a proton exchange membrane fuel cell (PEMFC) with an open-ended anode using agglomerate model: A CFD study A transient heat and mass transfer CFD simulation for proton exchange membrane fuel cells (PEMFC) with a dead-ended anode channelNatural-forced cooling and Monte-Carlo multi-objective optimization of mechanical and thermal characteristics of a bipolar plate for use in a proton exchange membrane fuel cell

·             The research in this paper is very innovative, but the practical significance of the project is not clear. I hope the author has an introduction in this regard

·             The authors should re-write the introduction to show the importance of the present work.

·             Based on several issues observed in this work, it is recommended that the authors read carefully and consider the following papers to resolve the above issues. 

·              

·             A good abstract should have the following elements (a) Introduction/ Background (b) Method (c) Results (d) Discussion/ Conclusion.

·             Please improve the language used in the present manuscript. Avoid long sentence.

·             Please improve the language used in the manuscript. Check entire manuscript

Reviewer 2 Report

In this manuscript, the authors use micro-electro-mechanical systems (MEMS) technology to develop a flexible micro-sensor to integrate temperature sensing, flow speed sensing and pressure sensing. This sensor can resist the high temperature electrochemical environment. The authors use it to monitor the working status of high   temperature proton exchange membrane fuel cells in real time. They test temperature, flow rate and pressure on the inlet, middle and outlet of fuel cells working at 160 ℃. Major revision is recommended.

1.      Does it affect each other when integrate the three sensors together? For example, the flow velocity sensor will produce a constant warmth field. Does this warm field affect the temperature sensor?

2.      Why are four devices (four red blocks) instead of 3 devices in fig. 1 (a) and (b)?

3.      The tests of sensors in fuel cells should be analyzed and evaluated. For example, the test data of the temperature sensor fluctuates very much in fig. 3. Can this reflect the real temperature in the fuel cell?

4.      What is the thickness of each layer prepared in the article? How thick are the electronics layer and the protective layer?

Reviewer 3 Report

See attachment for my feedback

Round 2

Reviewer 2 Report

The authors have solved all the questions. Thus I would recommend the acceptance.

Reviewer 3 Report

Although the authors did not address all questions (e.g., how the T-sensor is operated, in constant-T or constant-P mode etc), the manuscript is sufficiently improved to be accepted as communication.